# Detection of neutralizing antibodies against multiple SARS-CoV-2 strains in dried blood spots using cell-free PCR

Kenneth Danh[1], Donna Grace Karp[1], Malvika Singhal[1], Akshaya Tankasala[1], David Gebhart[1], Felipe de Jesus Cortez[1], Devangkumar Tandel[1], Peter V. Robinson[1], David Seftel[1], Mars Stone[2], Graham Simmons[2], Anil Bagri[3], Martin A. Schreiber[4], Andreas Buser [5], Andreas Holbro[5], Manuel Battegay[6], Mary Kate Morris[7], Carl Hanson[7], John R. Mills [8], Dane Granger[8], Elitza S. Theel [8], James R. Stubbs [8], Laurence M. Corash [3] & Cheng-ting Tsai [1✉]

An easily implementable serological assay to accurately detect severe acute respiratory syndrome coronavirus 2 (SARS-CoV-2) neutralizing antibodies is urgently needed to better track herd immunity, vaccine efficacy and vaccination rates. Herein, we report the Split-Oligonucleotide Neighboring Inhibition Assay (SONIA) which uses real-time qPCR to measure the ability of neutralizing antibodies to block binding between DNA-barcoded viral spike protein subunit 1 and the human angiotensin-converting enzyme 2 receptor protein. The SONIA neutralizing antibody assay using finger-prick dried blood spots displays 91–97% sensitivity and 100% specificity in comparison to the live-virus neutralization assays using matched serum specimens for multiple SARS-CoV-2 variants-of-concern. The multiplex version of this neutralizing antibody assay, using easily collectable finger-prick dried blood spots, can be a valuable tool to help reveal the impact of age, pre-existing health conditions, waning immunity, different vaccination schemes and the emergence of new variants-of-concern.

[1] Enable Biosciences Inc, South San Francisco, CA, USA. [2] Vitalant Research Institute, San Francisco, CA, USA. [3] Cerus Corporation, Concord, CA, USA. [4] Department of Surgery, Oregon Health & Science University, Portland, OR, USA. [5] Regional Blood Transfusion Service, Swiss Red Cross, University Hospital Basel, University of Basel, Basel, Switzerland. [6] Division of Infectious Diseases & Hospital Epidemiology, University Hospital Basel, University of Basel, Basel, Switzerland. [7] California Department of Public Health, Richmond, CA, USA. [8] Department of Laboratory Medicine and Pathology, Mayo Clinic, Rochester, MN, USA. ✉email: jasontsai@enablebiosciences.com

The current epidemic of COVID-19 (novel coronavirus disease-2019) caused by SARS-CoV-2 has propagated globally at unprecedented speed[1–5]. It has led to more than 522 million confirmed infections worldwide and over 6.2 million deaths[1–5]. SARS-CoV-2 virus enters human cells via binding between the viral surface spike protein and the human ACE2 receptor[5]. Neutralizing antibodies (Nab) are capable of disrupting this interaction and have been shown to result in enhanced disease survival and reduced viral loads in swab specimens[3,4]. NAb can be found in patient specimens after natural infection, vaccination and/or receipt of convalescent plasma treatment. Monitoring of Nab after these events can provide useful information to both predict disease progression and confirm vaccination or treatment efficacy.

The virus plaque reduction neutralization test (PRNT) is the current gold standard assay for NAb[6]. However, PRNT's reliance on infectious SARS-CoV-2 virions limits the use of this potentially hazardous and time-consuming assay to relatively few well-resourced institutions equipped with biosafety level 3 (BSL3) laboratories. Modifications to the PRNT such as pseudovirus neutralization assays insert sections of the virus in question into benign viral targets to allow for a safer approximation of PRNT, but are still reliant on time consuming cell-based methods[6] and give results that do not always match those of live-virus PRNT assays[7]. ELISA and microbead-based methods have been reported, but they are either not multiplexable or may not be applicable to challenging sample types such as dried blood spots[8,9].

In this study we develop and validate an assay, termed SONIA (Fig. 1), to measure NAb using several cohorts of well-characterized specimens. This assay is inspired by our previous work of an ultrasensitive and highly specific assay method termed antibody detection by agglutination PCR (ADAP). The ADAP platform has been successfully applied to a wide variety of infections and autoimmune diseases[10–14]. Notably, we also present data on a multiplex version of the cell-free PCR assay to measure NAb against the alpha and delta SARS-CoV-2 variants in finger-prick dried blood spot specimens.

## Results

### Selection of antigens for the SONIA neutralization PCR assay to measure Nab.
The successful development of the NAb assay relies heavily on the proper choice of the antigens used. To that end, we first evaluated assay performance using the S1 portion of the spike protein versus the receptor binding domain (RBD) fragments of the S1 protein. We assayed two convalescent COVID-19 patient samples and four control specimens from healthy blood donors collected prior to the outbreak (Fig. 2). The COVID-19 samples had been analyzed using a cell-based pseudovirus neutralization assay[15,16] and confirmed to contain high titers of NAb. For both antigens, we observed no competition signals from the negative control specimens, and strong competition signals from the COVID-19 samples, indicating effective competition and neutralization of the S1-ACE2 interaction. Given the observation of much stronger signals in the S1 protein-based neutralization assay (Fig. 2), we chose to proceed with the S1 protein for further validation. A possible explanation for this observation is that a NAb may neutralize by binding an adjacent epitope on the S1 protein outside of the RBD but still block binding to ACE2 through steric hindrance. Indeed, several studies have shown that the N terminal domain of S1 (outside of RBD) was a critical neutralization antibody binding target[17–20]. The assay had intra- and inter-assay variations of 3.52%–13.17% and 6.64%–16.72% respectively (Supplementary Table 1 and 2).

### Validation of SONIA neutralizing assay with characterized patient samples.
To further validate the assay, we tested serum samples from 146 COVID-19 patients collected 12 days after symptom onset and control specimens collected prior to the outbreak from 43 patients from other viral/bacterial infections, including HCV, flavivirus and Lyme disease using the SONIA neutralization assay (Fig. 3 and Table 1). NAb were detected in 145 COVID-19 serum samples but not in any of the 43 non-COVID-19 serum samples.

Notably, the 146 COVID-19 serum samples found positive by the SONIA neutralization test had been tested with a pseudoviral neutralization test (VNT) at Mayo Clinic. All samples with titers above 1:160 in the VNT ($n = 114$) were also positive for NAb by the SONIA neutralization test. For 32 COVID-19 patient samples with low VNT titers of 1:80, 31 were positive for NAb in the PCR-based assay.

Apart from the qualitative agreement, it is of interest to explore the quantitative agreement between the two assays. Thus, we sought to analyze the signal correlation between the two assays. There were two background considerations for this analysis. First, the VNT titer is not continuous (such as titers of 1:80, 1:160, and 1:320). Secondly, the PCR-based assay uses qPCR as assay readout, such that the signals are logarithmic in nature. Therefore, we first averaged the NAb signals for samples with the same VNT titer, and plotted the SONIA assay signals against log-transformed VNT titers (Fig. 4). The result yielded a Pearson

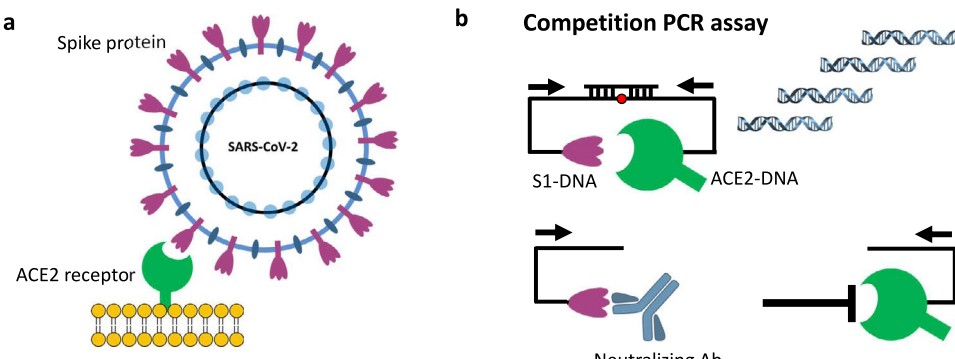

**Fig. 1 Principle of SONIA neutralization PCR test. a** Viral entry of SARS-CoV-2 is mediated by the binding of the spike protein to the human receptor angiotensin-converting enzyme 2 (ACE2). Disruption of this interaction forms the basis of neutralization by antibodies (NAb). **b** SONIA Neutralization PCR test reconstructs this interaction using a combination of S1 subunits of the spike protein- and ACE2-DNA conjugates. In the absence of NAb, S1 and ACE2 engage with strong affinity, thereby positioning the two DNA barcodes in proximity for subsequent ligation and PCR-amplification. On the other hand, binding of NAb blocks S1 subunit from binding ACE2, leaving the two DNA barcodes separated. Since each barcode has only one PCR primer binding site, they cannot be separately amplified. Therefore, the quantities of NAb are correlated with the decrease of PCR amplicon formation.

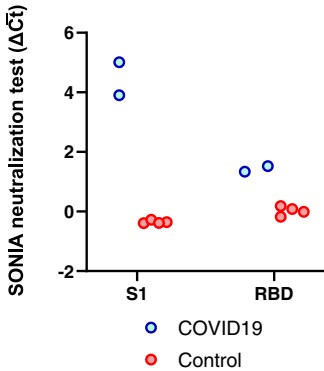

**Fig. 2 Selection of antigens for the SONIA neutralization PCR test.** Both S1 subunit and receptor binding domain (RBD) of spike protein have the ability to engage with the ACE2. To evaluate suitability for the neutralization PCR assay, we tested convalescent COVID-19 patient sera ($N = 2$, blue circle) and healthy donor sera ($N = 4$, red circle) collected prior to the outbreak. The COVID-19 sera were confirmed to harbor NAb by a pseudo-virus neutralization assay. The y-axis is the SONIA neutralization PCR test signal $\Delta\overline{Ct}$ calculated by subtracting the Ct value of the sample from that of a buffer only blank control.

correlation factor $R^2$ of 0.88 (95%CI: 0.75–1.02), indicating that the two assays were significantly correlated.

In summary, the validation above showed that the SONIA neutralization assay could detect NAbs with high sensitivity and specificity, and that the results were quantitatively and qualitatively concordant with standard VNT assays.

**Application of SONIA neutralization assays to investigate NAb development over time.** We sought to apply the SONIA neutralization assay to characterize NAb development over time in a cross-sectional patient cohort. We tested 5 samples from COVID-19 patients collected within 10 days post symptom onset, 5 samples between 10 and 12 days, 6 samples between 13 and 15 days, 5 samples between 16 and 18 days, and 7 samples 18 days after symptom onset. The NAb signals quickly increased between 10 and 12 days and reached a plateau 18 days after onset (Fig. 5). The kinetic profile observed was consistent with literature reports using VNT assays[21], providing an additional layer of evidence on a potential use case for the SONIA neutralization assay.

**Application of SONIA neutralization assay to detect NAb in at-home collected dried blood spot specimens.** The ability to detect NAbs in at-home collected dried blood spots might substantially enhance the reach, deployability and scalability of a NAb assay. To that end, we tested at-home collected dried blood spots from 27 PCR positive individuals and 20 healthy controls. These samples were collected for a previously reported study for COVID-19 total antibody detection[22]. The samples were collected by mailing collection kits to donors' homes and returned by USPS mail. All 27 PCR positive individuals were positive and all 20

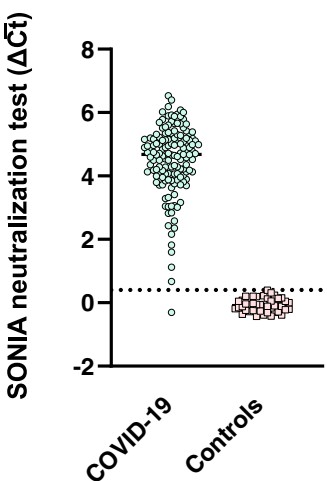

**Fig. 3 NAb reactivity in convalescent COVID-19 and control sera with non-COVID-19 infection by the SONIA neutralization PCR test.** Convalescent COVID-19 sera ($N = 144$) were from patients 14 days after symptom onset (blue circle), and control sera ($N = 43$) from patients with HCV, flavivirus infection and Lyme disease (pink square). The y-axis is the SONIA neutralization PCR test signal $\Delta\overline{Ct}$ calculated by subtracting the Ct value of the sample from that of a buffer only blank control.

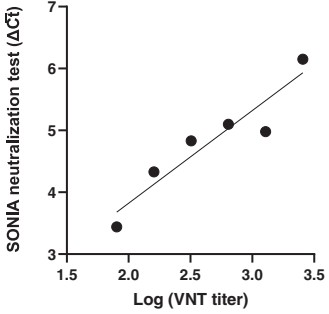

**Fig. 4 Correlation between the SONIA neutralization test and VNT titers.** The SONIA neutralization PCR test signals were averaged for samples with the same PRNT titers and plotted on the y-axis. Considering the PCR signals were logarithmic in nature, the VNT titers were log-transformed and plotted on the x-axis. The VNT was performed by the Mayo Clinic. The y-axis is the SONIA neutralization PCR test signal $\Delta\overline{Ct}$ calculated by subtracting the Ct value of the sample from that of a buffer only blank control.

**Table 1 Analysis of SONIA neutralization test with samples characterized by VNT ($N = 189$).**

| NAb titer by VNT | Number of samples | Positive by SONIA test | Average SONIA test signals | SD of SONIA test signals | Agreement |
|---|---|---|---|---|---|
| Pre-COVID | 43 | 0 | 0.07 | 0.22 | 100% |
| 80 | 32 | 31 | 3.44 | 1.34 | 97% |
| 160 | 34 | 34 | 4.33 | 0.76 | 100% |
| 320 | 37 | 37 | 4.83 | 0.91 | 100% |
| 640 | 26 | 26 | 5.10 | 0.96 | 100% |
| 1280 | 14 | 14 | 4.98 | 0.93 | 100% |
| 2560 | 3 | 3 | 6.15 | 0.26 | 100% |

Patient specimens were tested by SONIA and viral neutralization test (VNT). The concordance between the two methods was analyzed.

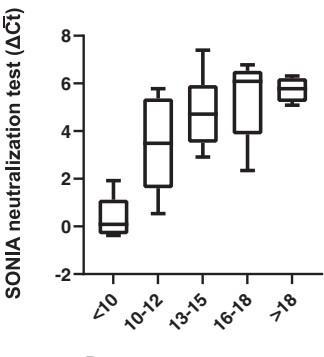

**Fig. 5 Evaluation of kinetics of NAb development by the SONIA neutralization PCR test.** Sera from COVID-19 patients after different days post symptom onset were tested by the neutralization PCR assay. All serum samples (N = 28) were from unique individuals. The y-axis is the SONIA neutralization PCR test signal $\Delta\overline{Ct}$ calculated by subtracting the Ct value of the sample from that of a buffer only blank control. The boxplots show medians (middle line) and 75th and 25th quartiles (upper and lower boxes), while the whiskers show the minimum and maximum of the data.

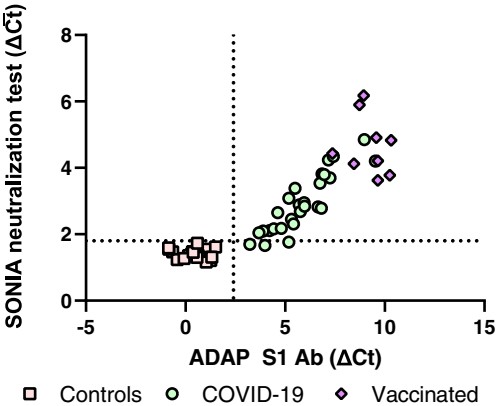

**Fig. 6 Analysis of neutralizing antibodies in at-home collected dried blood spot specimens.** Dried blood spot specimens from convalescent COVID-19 patients (n = 27, green circle), vaccinated (n = 9, purple diamond) and controls (n = 20, pink square) were analyzed by the SONIA neutralization PCR test for NAb. The y-axis is the SONIA neutralization PCR test signal $\Delta\overline{Ct}$ calculated by subtracting the Ct value of the sample from that of a buffer only blank control. The same samples were also analyzed for total spike (S1) antibodies with a previously reported method and plotted in the x-axis. The x-axis is the ADAP assay signals $\Delta Ct$ calculated by subtracting the buffer only blank control Ct from the Ct value of the sample[22].

healthy controls were negative for S1 antibodies. The analysis of NAb showed that 24 out of 27 PCR positive individuals had detectable levels of NAb, and that all 20 negative individuals were negative for NAb (Fig. 6). The figure also shows the correlation between S1 antibody level and NAb in dried blood spots. The 3 PCR-positive but NAb-negative individuals also had lower S1 Ab levels. Indeed, it has been reported that over 10% of convalescent patients do not harbor NAb within weeks of recovery[23–25]. Thus, it is possible that these individuals indeed did not harbor significant amounts of NAb in the first place, resulting in the observed negativity for NAb in dried blood spots.

To investigate if the SONIA neutralization test could be applied to monitor NAb resulting from vaccination, finger-prick dried blood spots were also collected from 9 donors at least 2 weeks after the first dosage of their mRNA 1273 vaccine. The dried

blood spots were analyzed for both total antibodies against S1 and NAb (Fig. 6). The data showed that substantial NAb signals were successfully detected for all vaccinated donors. Taken together, the data imply that at-home collected dried blood spots could be a promising matrix for large scale surveillance of NAb from both natural infection and vaccination.

**Validation of a multiplex SONIA neutralization assay for detection of NAb against variants-of-concern (VOC) of SARS-CoV-2 in finger-prick dried blood spots.** Since the beginning of the pandemic, SARS-CoV-2 virus has undergone rapid and frequent mutation. Certain mutated versions of the virus display enhanced transmission capability and varying degrees of evasion of the immunity induced by vaccination[9]. It is thus of great interest to develop an assay to measure antibody neutralizing capacity against each of these emerging VOC.

As an exploration of this feasibility, we obtained a supply of S1 subunits with amino acid sequences representing alpha variant B.1.1.7 and the delta variant B.1.617.2. The multiplex SONIA neutralization test was composed of the ACE2 receptor on a common DNA barcode, and the wild type (WT) S1, alpha S1 and delta S1 on unique DNA barcodes. In this manner, we can measure NAb against each of the S1 protein in a single assay by DNA barcoding.

We applied the 3-plex SONIA neutralization test to analyze finger-prick dried blood spot samples from 40 vaccinated individuals, 80 healthy controls, and 40 patients with Lyme disease (Fig. 7 and Table 2). Notably, matched serum samples from 40 vaccinated individuals were obtained and tested by live viral neutralization assay using wild-type and delta strain respectively, offering an opportunity to characterize the relative performance of the multiplex SONIA neutralization test against the gold standard.

For neutralizing antibody against wild type, all 120 non-COVID dried blood spots tested negative in the multiplex SONIA assay. Among the 40 vaccinated individuals, all of their serum samples had neutralizing antibody titers above 1:80 when tested by the wild type live virus assay, and 39 out of the 40 tested positive for wild type neutralizing antibodies using the multiplex SONIA assay.

For neutralizing antibodies against delta, all 120 non-COVID dried blood spots tested negative in the multiplex SONIA assay. Among the 40 vaccinated individuals, 34 of them had neutralizing antibodies of titer above 1:80 in serum samples when tested by the delta live virus assay, and 31 out of the 34 tested positive for delta neutralizing antibodies using the multiplex SONIA assay.

Taken together, the data showed that the multiplex SONIA assay had 91.2–97.5% sensitivity and 100% specificity against wild type and delta neutralizing antibodies in finger-prick dried blood spots. Notably, correlation analysis of observed SONIA signals in dried blood spots against the NAb titer seen in serum by live virus assays revealed these two assay types to be highly correlated (R = 0.92–0.93), indicating favorable quantitative agreement.

**Impact of specimen transportation and sample collection protocol deviation for the multiplex SONIA assay.** Despite the strong performance observed above, several outstanding questions remained to be addressed to ensure that the multiplex SONIA assay could be used in a real-world setting for the reliable and robust analysis of neutralizing antibodies.

First, the stability of the dried blood spot cards was evaluated by cycling the dried blood spot cards in temperature profiles mimicking typical summer and winter USPS shipping conditions (Supplementary Table 3 and 4). The results showed that the neutralizing antibodies were stable under potential temperature

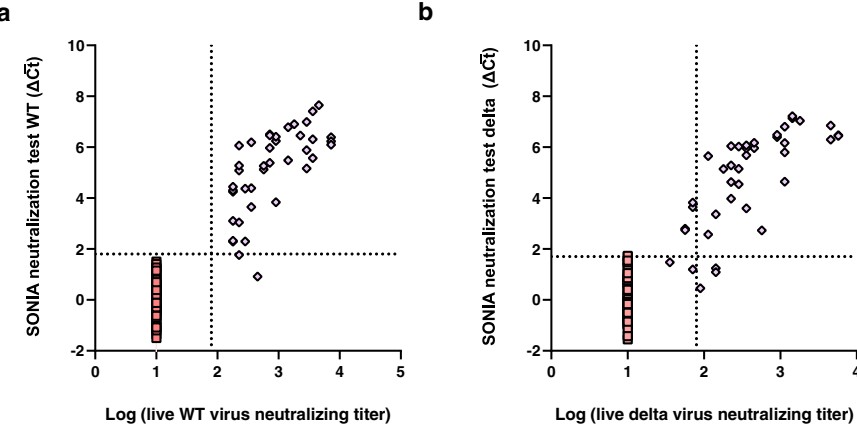

**Fig. 7 Multiplex SONIA neutralization test for analysis of NAb against variant of concern (VOC). a** Comparison of multiplex SONIA neutralization test for wild type against live wild type virus neutralizing assay (COVID-19 samples in purple diamond and controls in red square). **b** Comparison of multiplex SONIA neutralization test for delta against live delta virus neutralizing assay (COVID-19 samples in purple diamond and controls in red square). The vertical and horizontal dot lines are the cutoff of positivity for the live virus assay and the multiplex SONIA neutralization assay. The y-axis is the SONIA neutralization PCR test signal $\Delta \overline{Ct}$ calculated by subtracting the Ct value of the sample from that of a buffer only blank control.

**Table 2 Sensitivity and specificity of multiplex SONIA assay using dried blood spot specimens compared to live virus assays (N = 160).**

| | | Live virus assay positive | Multiplex SONIA assay positive | Agreement rate |
|---|---|---|---|---|
| Vaccinated group (N = 40) | Wild type | 40 | 39 | 39/40 (98%) |
| | Delta | 34 | 31 | 31/34 (91%) |
| | Alpha | — | 38 | — |
| | | Live virus assay negative | Multiplex SONIA assay negative | Agreement Rate |
| Healthy control group (N = 80) | Wild type | 80 | 80 | 80/80 (100%) |
| | Delta | 80 | 80 | 80/80 (100%) |
| | Alpha | 80 | 80 | 80/80 (100%) |
| Lyme disease group (N = 40) | Wild type | 40 | 40 | 40/40 (100%) |
| | Delta | 40 | 40 | 40/40 (100%) |
| | Alpha | 40 | 40 | 40/40 (100%) |

Matched serum and dried blood spot samples were obtained from the participants. Multiplex SONIA assay was performed on the dried blood spot specimens, and the live virus assays were performed for the wild type and delta strain for the vaccinated group.

fluctuations seen in the USPS shipping process (Supplementary Table 5).

Secondly, it is possible that a real-world user might deviate from the sample collection protocol such that the dried blood spot card may not be completely dry before being placed into a return USPS mailer box. We evaluated the impact of shorter drying times of 8 min, 30 min and 60 min, and showed minimal or no impact compared to the standard 240 min (4 h) drying protocol (Supplementary Table 6).

Thirdly, in the collection protocol, an alcohol cleaning pad was provided. It was of interest to evaluate if remaining alcohol on the fingertip might impact assay signals. To simulate a user wiping the subject's finger prior to obtaining the blood sample, pipette tips used to apply blood to the DBS paper were wiped with alcohol pads and allowed to dry for 4, 15, and 30 s. The results showed minimal or no impact on the assay integrity (Supplementary Table 7).

Finally, we evaluated the stability of dried blood spot cards and the eluents after freeze-thaw cycles to ensure that the samples could be accurately processed in an asynchronous batched manner. The results showed that the NAb against alpha, delta and wild-type (WT) strains remained intact after several freeze-thaw cycles. (Supplementary Table 8–13).

In summary, additional efforts were taken to evaluate impact from potential deviations during the specimen collection and

transportation process. The data demonstrated that the neutralizing antibody analytes were highly resistant to these deviations and supported dried blood spot as a reliable sample matrix for their detection.

## Discussion

Analytical techniques for detection of SARS-CoV-2 infection and immune status are essential for a variety of purposes. RNA-based tests detect actively infected patients who might be shedding infectious virus. Serological tests are required to measure the presence of antibodies to aid in understanding individual past exposure and community prevalence. Neutralizing antibody assays provide a closer link between immunity and potential clinical protection[26].

Interestingly, our study revealed that both the RBD and S1 proteins can be used as antigens for our neutralization assay. However, the S1 proteins showed stronger NAb signals. Past MERS studies have identified highly potent neutralizing monoclonal antibodies targeting regions outside of the RBD[17]. Similar findings were made recently for COVID-19, suggesting that assays using the S1 subunit or the full spike protein may generate more clinically relevant NAb signals[20]. It remains to be evaluated if SONIA neutralization assay signals using any of these antigens correlate with the often variable immune protection seen after SARS-Cov2 native infection.

It should be noted that the SONIA assay uses a chemical cross linker to install DNA barcodes on the protein. The resulting conjugates are ensembles of protein-DNA conjugates with different locations of DNA barcodes. This is beneficial for the detection of heterogenous human immune responses in that while some immunogenic epitopes may be masked in some of the probes, the same epitopes may be exposed in other probes. In addition, the chemical installation of DNA did not appear to undermine interactions between the S1 subunit and the ACE2 receptor, because neutralizing antibodies could be detected with high fidelity and concordance with label-free VNT assays.

Furthermore, the study showed that NAb signals are consistent with standard VNT and live virus neutralizing assays[21]. Application of the SONIA neutralization assay could also define kinetic profiles of NAb development consistent with literature reports. These data demonstrated that cell-free assay formats can obtain results that are concordant with cell-based assays.

Critical for ease of sample collection and scalability, this assay could be used to monitor NAb in dried blood spot specimens. All isotypes (e.g. IgG, IgM, IgA) are detected by SONIA. As vaccination becomes increasingly prevalent and repeated, it is of interest for both individuals and public health authorities to have reliable assurance that Nab are being successfully generated and maintained over time. Specifically it is important to track individuals longitudinally to characterize the longevity of NAb in populations. This is especially important in that there are several vaccination regimens available, and durability or waning responses may differ from one to another. Finally, countries such as Israel are requiring antibody tests as evidence of immune protection prior to entry. NAb could be an even more reliable marker for potential clinical protection for travel or close quarters congregation purposes.

A few studies have described surrogate blocking assays to detect neutralizing antibodies[8,9,27]. There is no FDA-authorized surrogate assay commercially available for measuring neutralizing antibodies in dried blood spots. The assay readout of the SONIA neutralization assay reported in this study is based on highly sensitive and commonly available qPCR instruments. The assay relies on the serial addition of key reagents, and does not require washing, centrifugation or cell culturing. The assay can be completed in as little as 140 min and is amenable to high-throughput implementation on laboratory liquid handling workstations. In particular, the throughput of this cell-free assay is markedly higher than cell culture-based traditional neutralizing assays. Furthermore, the assay leverages DNA barcoding to achieve multiplex detection. A detailed comparison chart between the SONIA and other neutralization tests is provided in supplementary materials. We showed that the NAb detected against the delta variant by multiplex SONIA neutralization testing were consistent with those found using a live delta virus neutralization assay. Importantly, the assay's robust compatibility with easy-to-collect and transport dried blood spot specimens could significantly lower the barrier to valued longitudinal testing in adults and children alike.

In conclusion, SONIA has the benefit of operational simplicity, rapid adaptability for new emerging variants and consistency with live virus neutralization assays. The wider deployment of this easy-to-use assay for the measurement of NAb can facilitate improved monitoring of vaccination rates and provide much needed community and personal guidance on the preferred type and timing of administered booster or variant-updated vaccines.

## Methods

**Ethical statement.** Blood specimens from SARS-CoV-2 RNA-positive individuals were obtained from various sources: Leftover serum samples originally submitted for COVID-19 serological testing were obtained from Oregon Health Sciences University Hospital (2 samples), the California Department of Public Health (25 samples), and the Mayo Clinic (143 samples). These were sourced from discarded clinical laboratory specimens exempted from informed consent and IRB approval under condition of patient anonymity, consistent with FDA guidance document FDA-2006-D-0095.

Three plasma samples were obtained from a previous study[28,29] as de-identified specimens. These samples were from COVID-19 convalescent outpatient plasma donors of the University Hospital Basel, Switzerland, obtained with informed consent and approved for use in research by the Ethics Committee of Northwest and Central Switzerland (IRB numbers Req-2020-00508 and EKNZ-2020-00769). The donors were screened in compliance with Swiss regulations on blood donation and approved as plasma donors in accordance with national regulations.

Convalescent COVID-19 patients and vaccinated individuals were newly recruited to donate matched serum samples and finger-prick blood for this study with informed consent under Enable Biosciences IRB #20180015 (approved by Western IRB).

Serum samples collected prior to the outbreak from blood donors and individuals with other viral or bacterial infection were purchased from commercial biobanks including 80 samples from BioIVT (Westbury, NY) and 40 samples from Boca Biolistics (Pompano Beach, FL). All specimens collected outside of Enable Biosciences clinical network were received as de-identified specimens.

**Materials.** The wild type SARS-CoV-2 S1 subunit, containing amino acids 1–674 with an Fc-tag at the C-terminus (#31806), expressed in HEK293 cells was purchased from the Native Antigen Company (Oxford, United Kingdom). The alpha lineage B.1.1.7 (alpha) S1 protein (#40591-V08H12) and delta lineage B.1.617.2 (delta) S1 protein (#40591-V08H23) with His-tag at the C-terminus were purchased from Sino Biologics (Beijing, China). The SARS-CoV-2 spike protein receptor binding domain (RBD) containing amino acids 319–541 with an Fc-tag at the C-terminus (#40592-V02H) and the ACE2 protein containing amino acids 1–740 with an Fc-tag at the C-terminus (10108-H05H) expressed in HEK293 cells were also obtained from Sino Biologicals (Beijing, China). Oligonucleotides used in the study were custom ordered from Integrated DNA Technologies (Coralville, IA). Platinum Taq polymerase (#10966026), SYBR qPCR 2X master mix (#4385610), Dithiothreitol (DTT #202090), and sulfo-SMCC (#22122) were purchased from ThermoFisher (Waltham, MA). DNA ligase (#A8101) was purchased from LGC (Teddington, United Kingdom). Other reagents are detailed in the method sections as appropriate.

**Synthesis of protein-DNA conjugates for neutralization assay.** For wild type S1, alpha strain (B1.1.7) S1, delta strain (B1.617.2) S1, and RBD and ACE2-DNA conjugates, the proteins were buffer-exchanged in reaction buffers (55 mM sodium phosphate, 150 mM sodium chloride, 20 mM EDTA, pH 7.2) to make 1 mg/mL solutions. A 1 µL solution of 8 mM sulfo-SMCC was added to 10 µL of each protein solution. The reaction mixtures were incubated at room temperature for 2 h. Thiolated DNA was suspended in reaction buffers to 100 µM. A 3 µL solution of thiolated DNA solution and 4 µL of 100 mM solution of DTT were mixed to reduce dimerized thiolated DNA to monomer forms. The solution was then incubated at 37 °C for 1 h. The excess sulfo-SMCC in protein mixtures and DTT in thiolated-DNA were removed by 7 K MWCO Zeba spin columns (Thermo Fisher, Waltham, MA). DNA and primer sequences were provided in Supplementary Table 15. The thiolated DNA and protein solutions were then pooled and incubated overnight at 4 °C. Finally, protein–DNA conjugates were purified by 30 kDa MWCO filter (Millipore, Burlington, MA). Conjugate concentrations were determined by BCA assay (Thermo Fischer). Conjugation efficiencies were analyzed by SDS-PAGE and silver staining as described previously[10–12]. DNA-to-protein ratios of the conjugates were estimated by UV–vis absorption and typically fell in the range of 1-to-1 to 2-to-1 (Supplementary Fig. 1). Protein–DNA conjugates were stored at 4 °C for short-term usage or aliquoted for long-term storage at −80 °C.

**SONIA neutralization PCR assay.** Neutralizing antibodies against SARS-CoV-2 are predominantly directed against the receptor binding domain (RBD) of the spike protein where they act to disrupt its interaction with the ACE2 receptor on the human cell surface[21,30,31]. We thus sought to recreate this competition between the NAb and ACE2 receptor for binding to the spike protein in an in vitro assay. While it is possible to develop such assays using ELISA or other solid-phase assay approaches[8], the ACE2 receptor is highly conformational and difficult to express. To mimic in vivo interaction of S1 subunit and ACE2 receptor, we constructed a solution-phase assay wherein a full-length DNA barcode is split into two halves—one installed on the S1 subunit protein, and the other installed on the ACE2 receptor (Fig. 1). In the absence of NAb, the S1 subunit protein and the ACE2 receptor naturally engage, positioning the two DNA barcodes in proximity. Then, the addition of a DNA ligase reunites the two barcodes to a full-length DNA amplicon for amplification and quantification. In contrast, when present in a sample, NAb will bind onto the S1 subunit protein DNA conjugate and prevent its interaction with the ACE2 receptor conjugate. In this scenario, attenuation of signal is observed, since the two DNA barcodes cannot come into the proximity for ligation and amplification. Given that there is no need for washing or centrifugation to remove the DNA-barcoded probes, the entire assay is conducted in

the solution phase, thus preserving the native antigen conformation. This DNA-based assay readout is inspired in part by previous work in the literature[22,32].

The cell-free SONIA neutralization assay detects NAb by observing the disruption of the interaction between the S1 protein and the ACE2 receptor (Fig. 1). Briefly, 4 µL of blood sample (e.g. serum, plasma, or dried blood spot eluent) was incubated with 2 µL containing 1 femtomole of S1-DNA conjugate mixtures at 37 °C for 30 min. Then, 2 µL containing 1 femtomole of ACE2-DNA conjugate mixtures was added and incubated at 37 °C for another 30 min. The neutralizing antibodies in the specimen will engage with S1-DNA conjugate in step 1 to decrease S1-DNA binding with ACE2-DNA in step 2. To quantify the degree of competition, 115 µL of ligation mix (20 mM Tris, 50 mM KCl, 20 mM MgCl2, 20 mM DTT, 25 µM NAD, 0.025 U/µl ligase, and 100 nM connector oligonucleotide) was added and incubated at 30 °C for 15 min. Then, 25 µL of ligated solution was mixed with the PCR master mix that contained primer pairs and polymerase for amplification under standard thermocycling conditions (95 °C for 10 min, 95 °C for 15 s, 56 °C for 30 s, 13 cycles). These pre-amplified products were quantified in a 96-well qPCR plate. SYBR green-based qPCR was performed on a Bio-Rad CFX96 real-time PCR detection system (95 °C for 10 min, 95 °C for 30 s, 56 °C for 1 min, 40 cycles). As a result of competition, specimens harboring high quantities of NAb will have less amplifiable DNA, thus a weaker qPCR signal (higher Ct). In contrast, samples without neutralizing antibodies will benefit from the binding between S1 and ACE2 proteins to generate a large amount of DNA amplicons, thus a stronger qPCR signal (lower Ct).

Instead of using the common cycle threshold (Ct) as a readout, the SONIA neutralization PCR assay readout is defined as the Ct value of the actual sample minus the Ct of a blank control, abbreviated as $\Delta\overline{Ct}$ ($\Delta\overline{Ct} = Ct_{sample}-Ct_{control}$). Since $\Delta\overline{Ct}$ is based on Ct values, it has arbitrary unit (a.u.).

The magnitude of the readout is proportional to the loss of amplicon concentration in the PCR plate well, which in turn is proportional to the amount of neutralizing antibody present in the sample. The subtraction of Ct offers significant reproducibility since the subtraction of the blank control Ct and the sample Ct cancels out any potential drift across runs. It should be noted that Ct value is logarithmic in nature. For instance, when Ct value differ by 2, the amount of quantities differ by 4-fold ($2^2$), instead of 2-fold as it would have been in other assays with linear readout[33]. Thus, when comparing the SONIA assay to traditional immunoassay, it is generally required to log-transform the traditional assay readout.

A comparison of SONIA neutralization PCR assay to other neutralization assays are provided in Supplementary Table 14.

**Multiplex SONIA neutralization PCR assay.** The multiplex SONIA PCR assay is similar to the singleplex protocol described above. Briefly, 4 µL of blood sample (serum or plasma) was incubated with 2 µL containing 1 femtomole of S1-DNA conjugate mixtures from each of the variant of concern at 37 °C for 30 min. Then, 2 µL containing 1 femtomole of ACE2-DNA conjugate mixtures was added and incubated. 115 µL of ligation mixtures were added and incubated. Then, in the pre-amplification step, primers specific to each of the variant and ACE2 conjugates were added for PCR amplification. Finally, in the real-time quantitative PCR step, amplified product is added to each well of qPCR plates containing specific primer pairs to each variant of concern.

**Dried blood spot collection and processing.** The procedure to collect and process dried blood spot specimens has been reported previously[22]. Briefly, dried blood spot specimens were collected by self-collected finger-prick blood dried on a standard Whatman 903 protein saver card. The cards were mailed back to the laboratory by standard postal service (typically received within 2–5 days) and stored at 2–8 °C until analyzed. For the elution process, six 3 mm disc were punched from the dried blood spots and incubated with 1000 µL of elution buffer for 90 min at 37 °C for elution. Then, the eluent was concentrated with a 100 kDa molecular weight cut-off column (MWCO, Millipore) for 8 min at 14,000 rcf. The eluent was either tested immediately or stored at −80 °C for long term storage.

**Viral neutralization test (VNT).** The VNT assay at Mayo Clinic for neutralizing antibodies (NAb) is an in vitro pseudovirus neutralizing assay that measures the ability of patient sera to prevent replication-competent vesicular stomatitis virus (VSV) expressing the SARS-CoV-2 spike protein from infecting engineered cells in a 96-well plate format using a luciferase reporter protein as reported previously[16].

For the VNT assay conducted at the California Department of Public Health (CDPH) shown in Fig. 7, live SARS-CoV-2 viruses were isolated and passaged as below. Variants were obtained from two sources. WA-1/2020 was obtained from WRCEVA and B.1.617.2 was isolated from a patient nasopharyngeal (NP) swab. To isolate B.1.617.2, 200 µL of a NP swab sample from a COVID-19 patient that was previously sequence-identified was diluted 1:3 in PBS supplemented with 0.75% bovine serum albumin (BSA-PBS) and added to confluent Vero-81 cells in a T25 flask, and allowed to adsorb for 1 h. Then additional medium was added and the flask was incubated at 37 °C with 5% CO2 for 3–4 days with daily monitoring for CPE. When 50% CPE was detected, the contents were collected, clarified by centrifugation and stored at −80 °C as passage 0 stock. Passaged stocks of both B.1.617.2 and WA-1/2020 were made by inoculation of Vero-81 confluent T150

flasks with 1:10 diluted p0 stock, similarly monitored and harvested at approximately 50% cytopathogenic effect (CPE). The stock was re-sequenced and TCID50 obtained by titration. These viral stocks were then used to conduct live viral assays. Briefly, CPE endpoint neutralization assays were done following the limiting dilution model. Using p1 stocks of lineages B.1.617.2 and WA-1/2020. The sample was diluted 1:10 and heat inactivated at 56 °C for 30 min. Serial 2-fold dilution of samples were made in BSA-PBS. Sample dilutions were mixed with 100 TCID50 of each virus diluted in BSA-PBS at a 1:1 ratio and incubated for 1 h at 37 °C. Final sample dilutions in sample-virus mixture ranged from 1:20 to 1:2560. A 100 µL of the sample-virus mixtures were inoculated with Vero-81 cells in 96-well plates in triplicate and incubated in a 37 °C with 5% CO2 incubator. After incubation 150 µL of MEM containing 5% FCS was added to the wells and plates were incubated at 37 °C with 5% CO2 until consistent CPE was seen in the virus control (no neutralizing sample added) wells. Positive and negative controls were included as well as cell control wells and a viral back titration to verify TCID50 viral input. Individual wells were scored for CPE as having a binary outcome of 'infection' or 'no infection' and the IC50 was calculated using the Spearman-Karber method. All steps were done in a Biosafety Level 3 (BSL3) lab using approved protocols.

**Statistics & reproducibility.** PRISM (version 8.1.1) and XLSTAT software (version 2018.1) were used for data and statistical analysis. For the correlation analysis, viral neutralizing assay signals were logarithmically transformed. The use of logarithm was necessary as SONIA neutralizing test readout is a logarithmic parameter due to reliance on Ct value. Two-tailed P values with an alpha of 0.05 were used as the cutoff for significance. The lab staff analyzed the samples in coded and randomized manner. Results were only unblinded after testing was completed.

**Reporting summary.** Further information on research design is available in the Nature Research Reporting Summary linked to this article.

## Data availability

All data are within the paper and its supporting information/source data file. Source data are provided with this paper.

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

## Acknowledgements
We are sincerely grateful to all patients, donors and staff who participated in this study. The study is funded in part by NIH SBIR 2R44DK110005-02 and 2RAI141118-02 to K.D., D.K., M.S., A.T., D.T., F.J.C., D.G., P.V.R., D.S. and C.T.T at Enable Biosciences. The content is solely the responsibility of the authors and does not necessarily represent the official views of the NIH. The findings and conclusions in this article are those of the authors and do not necessarily represent the views or opinions of the California Department of Public Health of the California Health and Human Services Agency.

## Author contributions
K.D., D.K., M.S., A.T., D.G., F.J.C., D.T., M.K.M. conducted the experiments and data collection. P.V.R., D.S., M.S., G.S., A.B., M.S., A.B., A.H., M.B., C.H., J.M., D.G., T.E., J.S., L.C. participated in the design and revision of the manuscript. C.T.T. designed and wrote the manuscript.

## Competing interests
K.D., D.K., M.S., A.T., D.T., F.J.C., D.G., P.V.R., D.S., and C.T.T. were employed by Enable Biosciences. F.J.C., D.G., K.D., D.T., P.V.R., D.S., and C.T.T. are shareholders of Enable Biosciences. P.V.R. and C.T.T. are inventors of the ADAP patent US11149296B2 licensed from the University of California, Berkeley to Enable Biosciences. The ADAP and SONIA assay used in this study is a product in development. This does not alter our adherence to journal policies on sharing data and materials. A.B. and L.M.C. were employees and shareholders of Cerus Corporation, a manufacturer of a convalescent plasma inactivation system. The remaining authors declare no competing interests.
