## [Peer Review File · Nature Communications]

REVIEWER COMMENTS

Reviewer #1 (Remarks to the Author):

The work is thoroughly prepared and I see no flaws in it. It would be interesting to see a side-by-side comparison to another neutralization test that is based on a solid phase. If not possible to perform from DBS, then that is also of interest to note in the paper.

The method is based on the proximity ligation assay which should be cited (Fredriksson et al 2002 Nature Biotechnology).

Also, the sequences used to couple to the reagents and PCR primers are not described making it difficult to reproduce.

The data is sound and the method clearly works well.

Reviewer #2 (Remarks to the Author):

This manuscript reports on a proximity-based PCR assay as a surrogate approach to monitor neutralizing antibodies of SARS-CoV-2. Protein-oligonucleotide conjugates of SARS-CoV-2 S1 or RBD proteins and ACE2; the binding of these proteins brings the oligos into proximity, enabling PCR amplification. Such assays have the potential to assess the presence and persistence of neutralizing antibodies without the need for specialized cell-based methods, especially considering the compatibility of the assay with dried blood samples. Although the assay appears to work, and the effort to assess the impact of specimen collection and transport is commendable, there are several technical ambiguities that need to be clarified to allow for better assessment of the platform.

DNA was conjugated to proteins using Sulfo-SMCC, which is a heterobifunctional linker that will react with the thiol of modified DNA and will react indiscriminately with primary amines on the target proteins. What is the average number of oligos attached to each of the proteins? How does this likely heterogeneous distribution of oligonucleotides affect the binding and readout of the assay (e.g. binding, or differences in distance between potential oligo partners – especially when considering the large Fc-tag on many of the protein)? Were the binding affinities of modified proteins compared to the unmodified versions ('strong affinity' is mentioned in the manuscript, but no values are found)? These issues seem paramount as there are many lysine residues present in the RBD of WT and VOC (Nat Rev Microbiol 19, 409–424 (2021) and these may be modified by Sulfo-SMCC conjugation. It also seems somewhat contradictory that the authors suggest that the deposition of ACE2 on solid supports "could potentially denature or mask critical epitopes" (with no citations to support this statement).

It is not clear what form of the S1 protein domain of each of the Sars-Cov-2 variants was used in the SONIA assay. Figure 1 caption states that "interaction using a pair of S1 subunits of the spike protein-"; however, Figure 2 reports "full S1" was used.

There are several questions regarding the comparison between SONIA and VNT titers (Figure 4). It is not clear what a discontinuous VNT titer is, or why this poses a problem for comparison. The Nab signals were averaged, but only the mean is reported and not any deviation – please include the deviations in the report, as well as the reflected error in the value for the Pearson correlation. Were the same dilutions used for both the SONIA and VNT?

What are the units for the SONIA neutralization test? It appears to be the change Ct number (, based on description in the method section, but a change Ct of 1-2 seems minimal. Is this actually fold-change? What is the limit of detection of this assay in terms of antibody concentration or even dilution titer for positive samples? Please explain the meaning of the statement "the PCR based assay used qPCR as assay readout, such that the signals were logarithm in nature"?

What are the units for Figure 6 x-axis? Is there an explanation what a significant number of convalescent patients appear to have no/low neutralizing ability? Is this purely an artifact of antibody titer as it appears in the Figure?

Although other surrogate assays are cited, these are not compared to the presented assay. It is stated SONIA does not require washing, but the multiple processing steps, including several mixing and incubation steps, and 140 min run time, are non-trivial and comparison/contrasting to other platforms would be valuable.

Unfortunately, There are a significant number of grammatical and formatting errors (e.g. different abbreviations "COVID19 vs COVID-19" "ELISA vs Elisa"; "close proximity" is redundant, 'signals were logarithm' spelling and grammar, text wrapping on tables in SI). These are distracting and decrease the readability of the manuscript.

Reviewer #3 (Remarks to the Author):

1. A fairly concise and succinct study concerning a novel COVID19 detection methodology. Fairly easy to follow and understand. A few minor revisions would help as detailed above.

2. It is not clear from the manuscript what the high-throughput capacity is for this approach. With PCR and robotics it's possible to perform 5000 or more tests per day. With rapid tests and assembly line approaches it's easy to perform 2-3000 tests per day. How does this approach fit into these numbers?

3. Since the presence of Ab in sera requires infection or vaccination followed by a recovery period to achieve detectable Ab titres, how soon after infection/vaccination can Ab be detected by this approach?

4. Is there a biological correlation between the SONIA result and infection? That is, does Nab titre and the SONIA result correlate with viral protection?

5. Can the methodology detect whether IgM or IgG Ab are being made?

6. Lines 403-413 (Tables S6 and S7) are not really germane to the main thrust of this paper.

7. On Figures 2-7 please either use a quantitative value for the y-axis or explain in much more detail what the shown values mean.

Thank you for considering our previous manuscript submission entitled “Detection of neutralizing antibodies against multiple SARS-CoV-2 strains in finger-prick dried blood spots using Split-Oligonucleotide Neighboring Inhibition Assay (SONIA)”. We appreciate the reviewers’ thoughtful and detailed comments and we have revised the paper to address those concerns. The reviewers’ comments are enumerated in italics, with our responses below.

Reviewer #1

1) The work is thoroughly prepared and I see no flaws in it.

We appreciate reviewer #1’s kind remark and recognition.

2) It would be interesting to see a side-by-side comparison to another neutralization test that is based on a solid phase. If not possible to perform from DBS, then that is also of interest to note in the paper.

There is no commercially available DBS assay readily available to perform side-by-side comparison. We have amended the manuscript in the Discussion section to highlight this fact as suggested (Line 257-259).

3) The method is based on the proximity ligation assay which should be cited (Fredriksson et al 2002 Nature Biotechnology).

We have added the original PLA paper to the citation list (REF #28).

4) Also, the sequences used to couple to the reagents and PCR primers are not described making it difficult to reproduce.

We have added the DNA sequences and primer sequences to the Supplementary Table 11.

5) The data is sound and the method clearly works well.

We appreciate again reviewer #1’s positive feedback.

Reviewer #2

1) This manuscript reports on a proximity-based PCR assay as a surrogate approach to monitor neutralizing antibodies of SARS-CoV-2. Protein-oligonucleotide conjugates of SARS-CoV-2 S1 or RBD proteins and ACE2; the binding of these proteins brings the oligos into proximity, enabling PCR amplification. Such assays have the potential to assess the presence and persistence of neutralizing antibodies without the need for specialized cell-based methods, especially considering the compatibility of the assay with dried blood samples.

We thank reviewer #2 for the succinct summary of the technology and the manuscript.

2) Although the assay appears to work, and the effort to assess the impact of specimen collection and transport is commendable, there are several technical ambiguities that need to be clarified to allow for better assessment of the platform. DNA was conjugated to proteins using

Sulfo-SMCC, which is a heterobifunctional linker that will react with the thiol of modified DNA and will react indiscriminately with primary amines on the target proteins. What is the average number of oligos attached to each of the proteins?

The average number of oligos attached to each of the protein is in the range of 1:1 to 2:1 (1-2 DNA per protein). We have further highlight this in the Method section (Line 328-331). We have also added Supplementary Figure 1 to provide additional data in this regard.

3) How does this likely heterogeneous distribution of oligonucleotides affect the binding and readout of the assay (e.g. binding, or differences in distance between potential oligo partners – especially when considering the large 'Fc-tag on many of the protein)?

We thank reviewer for the very good question, and we address this as below. Firstly, majority of the probe attachment methods used in standard immunoassays would also yield heterogenous distribution of probes. For instance, conjugation of fluorescent probes or HRP to the COVID-19 RBD (used in the Genscript assay, Tan, et al Nature Biotech 2020) could also lead to heterogeneous distribution of reporting probes. Secondly, the heterogeneity is in fact beneficial for the assay. Given the DNA installation location is somewhat random, each of the probe would have somewhat different epitope exposure. Thereby, by employing a heterogenous probe population, the assay would have a better chance to capture all relevant antibodies, as human antibody responses are polyclonal and highly heterogenous. We have added this discussion to the Discussion section (Line 234-242).

REF #7 Tan, C.W., Chia, W.N., Qin, X. et al. A SARS-CoV-2 surrogate virus neutralization test based on antibody-mediated blockage of ACE2–spike protein–protein interaction. Nat Biotechnol 38, 1073–1078 (2020).

4) Were the binding affinities of modified proteins compared to the unmodified versions ('strong affinity' is mentioned in the manuscript, but no values are found)? These issues seem paramount as there are many lysine residues present in the RBD of WT and VOC (Nat Rev Microbiol 19, 409–424 (2021) and these may be modified by Sulfo-SMCC conjugation.

We thanked reviewer #2 for raising this critical point for discussion. We have removed the wording “strong” affinity to prevent confusion (Line 375). While we didn’t perform SRP measurement to measure the affinity Kd between the modified protein probes, the fact that the conjugated S1 and ACE2 protein could generate low Ct values in the qPCR experiments, indicating that the S1 and ACE2 protein could successfully engage with one another after DNA installation. Furthermore, despite this work is unique in a sense that DNA tag were installed onto the protein, previous reported surrogate assay also used chemical approach to install HRP probes onto the RBD protein (Tan et al, Nature Biotech 2020). A typical HRP conjugation process would involve activation of lysine, then react the activated protein with HRP for conjugation. This is consistent with this work that we employed a different chemical activator to activate the same amino residue within the spike protein. Therefore, this study and previous reported study (Tan et al, Nature Biotech 2020) have demonstrated such activation would not impact the binding between ACE2 and RBD.

REF #7 Tan, C.W., Chia, W.N., Qin, X. et al. A SARS-CoV-2 surrogate virus neutralization test based on antibody-mediated blockage of ACE2–spike protein–protein interaction. Nat Biotechnol 38, 1073–1078 (2020).

5) It also seems somewhat contradictory that the authors suggest that the deposition of ACE2 on solid supports “could potentially denature or mask critical epitopes” (with no citations to support this statement).

The remark on ACE2 denaturation or epitope masking is mentioned in the original manuscript in the Method section and is based on observation of behavior from other proteins on solid support. Considering this is of speculation in nature and no literature had confirmed this hypothesis, the authors had removed that statement to avoid any confusion (Line 341 to 344).

6) It is not clear what form of the S1 protein domain of each of the Sars-Cov-2 variants was used in the SONIA assay. Figure 1 caption states that “interaction using a pair of S1 subunits of the spike protein-”; however, Figure 2 reports “full S1” was used.

We thanked the reviewer for capturing this. The S1 subunit were referred to “S1” as a subunit within the full spike protein, which is composed of S1 and S2. The full S1 is referring to the S1 subunit. In avoidance of doubt, we have standardized the naming to S1 subunit to better reflect it is S1 subunit within the spike protein.

7) There are several questions regarding the comparison between SONIA and VNT titers (Figure 4). It is not clear what a discontinuous VNT titer is, or why this poses a problem for comparison. The Nab signals were averaged, but only the mean is reported and not any deviation – please include the deviations in the report, as well as the reflected error in the value for the Pearson correlation. Where the same dilutions used for both the SONIA and VNT?

VNT or viral neutralization test readout is based on serial dilution of a sample until it can no longer inhibit viral growth to a pre-set threshold. The highest possible dilution is then defined as the VNT titer. For example, the readout is titer of 80, 160, 320, 640. Therefore, two samples with titer of 80 could harbor different quantities of neutralizing antibodies because titer of 80 means they have titer above 80 but below 160, but the precise titer could be anywhere within that range. On the contrary, the SONIA assay test the sample as it is without dilution, and generate a numerical Ct value readout, which is a continuous variable. In order to cross-compare these two distinct methods, it is necessary to pool the signals from all of the sample with the same titer together. Despite each sample with titer of 80 could have different true antibody levels, the average of them should generate a representative signals of that titer class.

Nevertheless, we acknowledge the reviewer’s feedback and thus we had added the distribution of signals for each titer class in the Table I of the manuscript, and we also reported the range of the Pearson correlation value ($R^2=0.88$, 95%CI 0.75-1.02) (Line 111). The same sample sets are tested by SONIA and VNT as the reviewer correctly pointed out.

8) What are the units for the SONIA neutralization test? It appears to be the change Ct number, based on description in the method section, but a change Ct of 1-2 seems minimal. Is this actually fold-change? What is the limit of detection of this assay in terms of antibody concentration or even dilution titer for positive samples? Please explain the meaning of the

statement “the PCR based assay used qPCR as assay readout, such that the signals were logarithm in nature”?

The SONIA assay uses qPCR as assay readout. In a standard qPCR experiment (such as those for COVID-19 RNA detection), the qPCR readout would be Ct values (e.g. Ct values of 25, Ct values of 35). The Ct value is defined as the number of PCR cycles needed for a sample to generate signals above a pre-defined fluorescent threshold. Therefore, a change of Ct value of 1 actually means it takes one more PCR amplification cycle to reach the same fluorescent signal. Considering each PCR cycle amplifies 1 molecule to 2 molecules, 2 molecules to 4 molecules, a change of Ct value by 1 means the analyte concentration differ by 2 fold. And a change of Ct values by 2 means the analyte concentration differ by $2^2=4$ -fold. Thus, in order to compare SONIA assay to other assay with linear readout (e.g. signals increase 2 times when there is twice as much analyte), we need to log transform their signals. We have added additional paragraph in the method section to elaborate this concept (Line 386-390). Since the SONIA readout is based on Ct values, it is unitless. It is difficult to define a meaningful limit of detection for neutralizing assay. For instance, we can attempt to determine LOD by obtaining a monoclonal antibodies with neutralizing capacity, and test the limit of detection by serially dilute the monoclonal antibodies. Nevertheless, unless we also evaluated other assay platform using the exact monoclonal antibodies, the limit of detection won't be cross-comparable across platform, because different monoclonal antibodies could have different binding affinity and different binding epitopes. An assay may have lower limit of detection when challenged with one monoclonal antibody, but have higher limit of detection when tested by another. The limit of detection could be addressed when a set of internationally recognized monoclonal neutralizing antibodies source become widely available.

9) What are the units for Figure 6 x-axis? Is there an explanation what a significant number of convalescent patients appear to have no/low neutralizing ability? Is this purely an artifact of antibody titer as it appears in the Figure?

The unit of the x-axis on Figure 6 is arbitrary unit (a.u), as previously reported in Karp, et al Scientific Report 2020. We have added additional citations to support the observation that 3 out of 27 convalescent patients (~11%) had no neutralizing antibodies. The cited publication reported 10-40% of convalescent patients lack neutralizing antibodies (titer <80), supporting the observation made in the manuscript. We have added the following citations to the manuscript.

REF #30 Edara VV, Hudson WH, Xie X, Ahmed R, Suthar MS. Neutralizing Antibodies Against SARS-CoV-2 Variants After Infection and Vaccination. *JAMA*. 2021;325(18):1896–1898.

REF #31 Zeng C, Evans JP, Pearson R, Qu P, Zheng YM, Robinson RT, Hall-Stoodley L, Yount J, Pannu S, Mallampalli RK, Saif L, Oltz E, Lozanski G, Liu SL. Neutralizing antibody against SARS-CoV-2 spike in COVID-19 patients, health care workers, and convalescent plasma donors. *JCI Insight*. 2020 19;5(22):e143213.

REF #32 Dispinseri S, Secchi M, Pirillo MF, Tolazzi M, Borghi M, Brigatti C, De Angelis ML, Baratella M, Bazzigaluppi E, Venturi G, Sironi F, Canitano A, Marzinotto I, Tresoldi C, Ciceri F, Piemonti L, Negri D, Cara A, Lampasona V, Scarlatti G. Neutralizing antibody responses to SARS-CoV-2 in symptomatic COVID-19 is persistent and critical for survival. *Nat Commun*. 2021 11;12(1):2670.

10) Although other surrogate assays are cited, these are not compared to the presented assay. It is stated SONIA does not require washing, but the multiple processing steps, including several mixing and incubation steps, and 140 min run time, are non-trivial and comparison/contrasting to other platforms would be valuable.

We have added a comparison Table against PRTN and surrogate ELISA in the supplementary section to help consolidate the pros and cons of each platform (Supplementary Table 10).

11) Unfortunately, There are a significant number of grammatical and formatting errors (e.g. different abbreviations “COVID19 vs COVID-19” “ELISA vs Elisa”; “close proximity” is redundant, ‘signals were logarithm’ spelling and grammar, text wrapping on tables in SI). These are distracting and decrease the readability of the manuscript.

We thanked the reviewer for identifying the grammatical and formatting errors. We have revised the manuscript to ensure consistency throughout the manuscript.

Reviewer #3

1) A fairly concise and succinct study concerning a novel COVID19 detection methodology. Fairly easy to follow and understand. A few minor revisions would help as detailed above.

We thanked reviewer #3 for the rewarding remarks.

2) It is not clear from the manuscript what the high-throughput capacity is for this approach. With PCR and robotics its possible to perform 5000 or more tests per day. With rapid tests and assembly line approaches its easy to perform 2-3000 tests per day. How does this approach fit into these numbers?

The high-throughput nature of the SONIA assay is compared to the gold standard live virus assay, which takes days to weeks to complete a batch of testing. While rapid test can be easily done, to the best of author’s knowledge, there is no rapid test for neutralizing antibodies. Most of the readily available rapid test pipeline is for generic binding antibodies, but not neutralizing antibodies. We have amended the manuscript to emphasize the comparison to the live virus assay to avoid any confusion in the Discussion section (Line 264-266).

3) Since the presence of Ab in sera requires infection or vaccination followed by a recovery period to achieve detectable Ab titres, how soon after infection/vaccination can Ab be detected by this approach?

We thanked reviewer to raise this critical question. In the Figure 5, we evaluated the kinetic of neutralizing antibody generation. Based on that data, we estimated it took 2 weeks after infection for neutralizing antibody to be detected.

4) Is there a biological correlation between the SONIA result and infection? That is, doe Nab titre and the SONIA result correlate with viral protection?

The correlation of neutralizing antibodies to viral protection had been reported in several literature. Considering the fact that the SONIA assay had high correlation with gold standard

NAb assay, the author believes SONIA assay results are also correlated with protection. Nevertheless, the author are procuring longitudinal patients cohorts to evaluate this hypothesis. The author had mentioned in the original manuscript to point out this important ongoing work (Line 231 to 233).

5) Can the methodology detect whether IgM or IgG Ab are being made?

The methodology is isotype independent. Therefore, the SONIA assay detects all isotypes of neutralizing antibodies. In order to detect specific isotypes, a sample treatment step is needed to isolate IgM or IgG for focus detection. We have added a clarification in the Discussion section (Line 249).

6) Lines 403-413 (Tables S6 and S7) are not really germane to the main thrust of this paper.

We thanked reviewer to point this out, and we agreed with the reviewer that those two data are not critical to the validity of the assay. The additional investigations in S6 and S7 are meant to demonstrate the real-world practicability of the method. While being of remote relevance, the investigation of S6 and S7 may address questions surrounding impact of potential contamination on the SONIA assay. Considering we are using qPCR as readout without any sample purification step, the author thinks it might be beneficial to retain those data in the manuscript but only in the supplementary table.

7) On Figures 2-7 please either use a quantitative value for the y-axis or explain in much more detail what the shown values means.

We have further amended the figure legend to clearly indicate the value in y-axis is based on Ct values differences, thus being an arbitrary unit.

We sincerely thank the reviewers again for their constructive feedback that has helped substantially augment the quality of the manuscript.

Sincerely,

Cheng-ting Tsai, PhD

REVIEWERS' COMMENTS

Reviewer #2 (Remarks to the Author):

The responses to concerns are detailed and comprehensive. The authors have provided additional data and explanations that improve the clarity of the manuscript.

Reviewer #3 (Remarks to the Author):

The authors have sufficiently responded to my criticisms and I am in agreement that it can now be accepted for publication.